# The Association of Maternal BMI with Overweight among Children Aged 0–59 Months in Kenya: A Nationwide Cross-Sectional Study

**DOI:** 10.3390/ijerph20021413

**Published:** 2023-01-12

**Authors:** Amos Mulu, Subas Neupane

**Affiliations:** Unit of Health Sciences, Faculty of Social Sciences, Tampere University, 33100 Tampere, Finland

**Keywords:** childhood, overweight, obesity, body mass index (BMI), maternal BMI, demographic and health survey, Kenya

## Abstract

Childhood overweight is a growing global public health challenge and is prevalent in many countries. We aimed at exploring the prevalence of childhood overweight and the association of maternal body mass index (BMI), maternal demographic factors, and child-related factors with childhood overweight among Kenyan children aged 0-59 months. This study utilized Kenya’s 2014 demographic and health survey, which was based on national representative cross-sectional data. A total of 8316 children and their mothers’ data were analyzed. Overweight in children and maternal BMI were defined using WHO standard criteria. Multivariate logistics regression models were used to study the association of maternal BMI and childhood overweight. Nationally 5% of Kenyan children aged 0–59 months are overweight (5.5% male vs. 3.8% female). The highest prevalence in overweight among children was found in Central region (6.9%) and lowest in North Eastern (3.1%) which could be explained by the various economic disparities. Maternal BMI with underweight was associated with lower odds (OR 0.30, 95% CI 0.14–0.64) whereas, overweight and obesity were associated with higher odds of overweight (OR for overweight 1.64, 95% CI 1.28–2.11 and OR for obesity 1.74, 95% CI 1.22–2.47) among their children compared to normal weight mothers. Overweight among children is of great concern and therefore initiatives to tackle both child and maternal health are urgently needed.

## 1. Introduction

The World Health Organization (WHO) stated that obesity has tripled since the year 1975 globally and 1.9 million adults became overweight and 650 million obese in 2016 [1]. In addition, 340 million children and adolescents aged between 5–19 years were either overweight or obese in 2016 [1]. Overweight (body mass index 25–29.9 kg/m^2^) and obesity (BMI ≥ 30 kg/m^2^) are the leading causes of almost 4.7 million premature deaths in the world [2]. In high-income countries such as the USA, almost two in three adults are either overweight or obese, and in south Asia and sub-Saharan Africa one in five adults have been reported to be overweight [2]. In the year 2020, 39 million children under the age of 5 years were either overweight or obese globally, half of whom are said to be from Asia and a quarter from Africa [1]. Childhood overweight and obesity is a great challenge in many countries, especially to those populations living in urban areas [3].

Kenya is in an epidemiological transition and has experienced a large increase in non-communicable diseases, obesity, conditions associated with urbanization, and less active lifestyles combined with re-emerging infectious diseases such as HIV/AIDS and tuberculosis [4]. However, according to the 2020 global nutrition report, Kenya is on course to meet at least four of the global nutritional targets for under-five overweight, stunting, wasting, and infant exclusive breastfeeding [5]. Earlier studies from Kenya, which examined geographic variations in the nutritional status of mothers and children, showed that the majority of the overweight population lived in Nairobi, and the underweight population are scattered in the rural areas of Kenya [6]. This explains how the area of residence could be influential when it comes to nutritional issues. In fact, some studies have found that there is a significantly higher percentage of children with high BMI living in urban areas [7]

Increase in maternal BMI is directly associated with childhood overweight and obesity [8]. High maternal BMI before pregnancy and gestational weight gain increases the risk of childhood overweight [9,10]. Once considered a high-income country problem, overweight and obesity are now rising amongst low- and middle-income countries, particularly in urban settings. Early childhood overweight and obesity affects both the physical and psychological health of children and may put them at risk of ill health in their adulthood. A study from Australia found that those who experienced overweight or obesity had higher risks of cardio-metabolic risk, higher risks of non-alcoholic fatty liver, as well as experiencing more negative psychological outcomes such as depression and low self-esteem when compared to those with normal weight [11]. Early childhood obesity has a great impact on children’s physical health, social and emotional well-being, self-esteem issues, and is also associated with poor academic performance [12]. It is also suggested that poor health stemming from childhood overweight/obesity can continue into adulthood [1]

Maternal nutrition has emerged as a key determinant of children’s nutritional health. The odds of children experiencing obesity increase tremendously when the mothers have preconception obesity [8]. A study by Gewa found that maternal obesity was strongly associated with higher odds of overweight and obesity among Kenyan children [7]. In addition, a study conducted in 34 sub-Saharan African countries associated maternal obesity with increased odds of neonatal deaths and more especially during the first week of neonatal life [13]. The current study aimed at ascertaining the prevalence of child overweight and analysing the association of maternal BMI, sociodemographic characteristics of the mother (age, level of education, residence, region of residence, and wealth quantile), and child-related factors (age, breastfeeding, and gender) with overweight among children aged 0–59 months, which had not been explored in Kenyan children by the previous studies. The vital evidence generated from the study will be of great importance in for stakeholders and policy makers in developing effective strategies to tackle childhood overweight in Kenya. 

## 2. Materials and Methods

### 2.1. Study Design and Data Source

The current study used data from the 2014 Kenya demographic health survey (KDHS). The data were collected during May–October 2014. The sample was drawn from the Fifth National Sample Survey and Evaluation Programme of the national bureau of statistics. The sample was designed to comprise 40,300 households from 1612 clusters (995 clusters from rural) spread across the country. Samples were selected independently in each sampling stratum, using a two-stage sample design. These clusters were selected with equal probability from the Fifth National Sample Survey and Evaluation Programme at the first stage. Then households from listing operations served as the sampling frame for the second stage of selection, whereby 25 households were selected from each cluster. The interviewers visited only the preselected households, and no replacement of the preselected households was allowed during data collection. The Household Questionnaire and the Woman’s Questionnaire were administered in all households, and the Man’s Questionnaire was administered in every second household.

A total of 21,435 (unweighted) children under the age of 5 and 13,143 (unweighted) mothers were eligible for weight and height measurements. However, because of the non-proportional allocation to the sampling strata and the fixed sample size per cluster, the survey was not self-weighting. Therefore, for this study, data were weighted to be representative at both national and regional levels. This study includes data from children-mother pairs (*n* = 8316) whose weight and height were measured and recorded.

Measurements of height and weight were obtained for all children born since January 2009 and listed in the Household Questionnaire. Interviewers carried a measuring board and electronic SECA scales with a digital screen. The measurement tools were designed and manufactured under the authority of the United Nations Children’s Fund (UNICEF). The scale allowed weighing of very young children through an automatic mother-child adjustment that eliminated the mother’s weight while she was standing on the scale with her baby. Weight was measured to the nearest 100 grams (g). Height measurements were made using height/length boards. Children less than 24 months or less than 78cm were measured lying down on the board (recumbent length). Heights for older children were measured while they were standing. 

### 2.2. Measurement of Variables

Overweight among children: Following WHO guidelines, the overweight among children was defined as weight-for-height z-score greater than two standard deviations above WHO Child Growth Standards median based on the height and weight [14]. All others with weight-for-height z-score, less or equal to two standard deviations was categorized as normal or underweight.

#### 2.2.1. Maternal/Parental Related Factors 

*Mother’s BMI*: Mother’s BMI was categorized using the WHO standard categorization into four categories: Underweight ≤18.49 kg/m^2^, normal weight 18.50–24.99 kg/m^2^, Overweight 25.00–29.99 kg/m^2^ and Obesity ≥30.00 kg/m^2^ [15]

*Mother’s age in years*: mother’s age was categorized into 5-year groups as follows: 15–19, 20–24, 25–29, 30–34, 35–39, and 40–49. 

*Type of residence*: Residence of mother was categorized into either urban or rural. 

*Region*: There are eight geographical regions in Kenya namely, North Eastern, Eastern, Coast, Central, Rift Valley, Western, Nyanza and Nairobi. This categorization aids in comparing the nutritional status of the residents of the various regions in order to identify intervention needs. 

*Level of education*: the education levels of both the mother and father/partner were determined as no education, primary, secondary, and higher level. 

*Quintile of wealth index*: The household wealth index was used as a proxy for a household’s long-term standard of living. For this study, wealth quintiles were expressed in terms of quintiles of individuals in the overall population rather than quintiles of population indicator. The wealth quantile index was ranked into five categories: lowest, lower, middle, higher, and highest. This was done through assigning a household score to each de jure household member, ranking population members by their score, and then dividing the ranking into the five equal categories, each comprising 20% of the population.

*Number of children in household:* The number of children who were under the age of 59 months in each participation household was taken into consideration and categorized into six groups which indicated the number of children in a household: 1, 2, 3, 4, 5, and equal to or above 6.

#### 2.2.2. Child-Related Factors

Child’s gender: (male/female), children’s age categorized in months as follows: 0–11, 12–23, 24–35, 36–47, 48–59 months, and lastly, breastfeeding period in months where the periods were categorized as: ≤12 months, 12–24 months or ≥25 months).

### 2.3. Statistical Analysis 

We used SPSS version 26 for the statistical analysis. We used a sampling strategy by using sample weights to estimate the distribution of independent and dependent variables. A descriptive analysis of the study sample was conducted and presented as frequency and percentages. Chi-square test was used to study the difference in maternal and child-related factors of the studied population according to the outcome and maternal BMI. Bivariate and multivariate logistics regression was used to study the association of the independent variables with the outcome. Odds ratio (OR) and their 95% confidence intervals (CIs) for the relationship between overweight of children and selected variables was calculated. The associations were set to be statistically significant at a *p* < 0.05. Three models were fitted; Model I examined bivariate (crude) association, Model II examined the multivariable model adjusted for maternal/parental factors, and Model III examined the multivariable model adjusted for maternal/parental and child-related factors.

### 2.4. Ethical Statement

The ICF Institutional Review Board and the National Ethics Committee of Kenya approved the survey procedures and instruments of the KDHS 2014. The Kenya National Bureau of Statistics (KNBS) conducted the 2014 KDHS survey including planning, analysis, and dissemination of the survey results. As the implementing agency, KNBS coordinated operational matters including planning, conducting fieldwork and processing collected data. Before interview, an informed consent statement was read to the respondent, who had option to either accept or decline to participate. A parent or guardian provided consent prior to participation by a child. In addition, for this study permission to use the KDHS data was obtained from the demographic health survey Program.

## 3. Results

Demographic characteristics of mother and child-related factors of the studied population are presented in Table 1. Among the factors for the mother, 21.3% and 9.0% were overweight and obese respectively. A majority of mothers (30.8%) were of age group 25–29 years, two thirds lived in urban areas and 29.0% lived in Rift Valley. 56% of mothers had attained a primary level of education, 51% of the partners had primary level of education and about 24% of the studied population were in the lowest quintile of the wealth index. Among child-related factors, 22.7% mothers had two children, 21% had children of age 13–24 months, and half of them were males. 38.8% and 51.6% of the children were breastfed for 12 and 24 months consecutively with 9.6% being breastfed for more than 25 months. 

The association of maternal characteristics and child-related factors with child’s BMI from chi-square text is presented in Table 2. Children from mothers experiencing overweight and obese mothers had the highest prevalence of overweight (7.7% and 7.9% respectively). The prevalence of child overweight increases as the mother’s BMI increases. Children of mothers in the age groups 15–19 and 25–29 years had the highest prevalence of overweight (5.6% and 5.4% respectively). The prevalence decreased with increasing age with lowest prevalence among mothers 40–49 years old. Significantly higher prevalence was found among urban residents compared to their rural counterparts (6.1% vs. 3.9%). Mothers with higher education had more overweight children when compared to those with no education (9.0% vs. 2.1%). Likewise, children from partners with higher education had the highest prevalence of overweight (8.7%). The prevalence of child overweight increases as the parental education level increases. Significantly higher prevalence of child overweight was found among households with one child compared to the household with more than six children (6.7% vs. 2.9%) The fewer the number of children in a household the higher the prevalence of child overweight. More boys (5.5%) were found to be overweight compared to girls (3.8%). Similarly, overweight in children was significantly high among 12 months’ old (7.9%) children compared to children at 59 months’ old (1.8%). The prevalence of child overweight decreases as the child’s age increases. Children breastfed for only 12 months had significantly higher prevalence of overweight (6.4%) compared to those breastfed for 24 months or more.

The association of maternal BMI with child overweight is presented in Table 3. The result shows that, underweight, overweight and obesity of mothers were associated with overweight among their children compared to normal weight mothers. Children of underweight mothers were 70% less likely to experience overweight compared to the children of mothers who were of normal weight. This remains consistent when all maternal and child-related factors were simultaneously adjusted in Model III (OR 0.30, 95% CI 0.14–0.64). Whereas children born from mothers with overweight and obesity were almost two-fold more likely to have overweight children compared to normal weight mothers. This remains significant when all maternal demographic characteristics were adjusted in Model II. In Model II when child-related factors were added in the model, the significant association still retained (OR for overweight 1.64, 95% CI 1.28–2.11 and for obese 1.74, 95% CI 1.22–2.47). Mothers in the age groups 25-29 (OR = 1.34 95% CI 0.71–2.21) and 30–34 (OR = 1.18 95% CI 0.60–2.35) years were associated with overweight experience among their children compared to their counterparts in the age group 15–19 years. Age groups 20–24 and 40–49 years were associated with lower odds of childhood overweight in all the three models. 

Compared to rural residence, urban residence was significantly associated with overweight experience among children in the crude model. However, the significant association was lost when maternal and child-related factors were added in the model. Compared to the North Eastern region, children of mothers living in Nairobi (OR = 2.02, 95% CI 1.23–3.34), Central (OR = 1.86, 95% CI 1.11–3.11) and Western region (OR = 1.75, 95% CI 1.04–2.93) had higher odds of experiencing overweight in the final model. The less the parent is educated the lower the odds of experiencing an overweight child. However, the association for both mother’s and partner’s education were significant in the crude model only. 

Compared to mothers in the lowest quintile of the wealth index, mothers in upper wealth quintiles had increased odds of having their child overweight. The association follows the dose response manner in the crude model. However, when maternal factors and child-related factors were added in the models (Model II and Model III), the association attenuated significantly. Compared to mothers with one child, those having more children ever born had lower odds of having overweight children. However, this association was significant only in the crude model. 

Among child-related factors, male gender had higher odds of overweight when compared to female (OR = 1.53, 95% CI 1.24–1.90). Compared to 59 months old, younger age children had significantly higher odds of being overweight. The association follows a dose response manner. However, the association reversed with still retained statistical significance and the dose response manner in Model III. The breastfeeding period was associated with child overweight. When compared to children who were breastfed for 12 or less months, those breastfed for 12–24 months had lower odds of experiencing overweight children (OR = 0.70, 95% CI 0.57–0.86) in the crude model. However, this association attenuated in the full model (Model III), whereas those who were breastfed for 25 or more months had 75% lesser likelihood of having overweight children. This remains significant even in the final model with 49% lower odds of having overweight children (OR = 0.51, 95% CI 0.28–0.95). 

## 4. Discussion

The aim of this study was to ascertain the prevalence and study the association of maternal BMI with overweight children adjusted for various maternal and child-related factors among children aged 0–59 months in Kenya. The national prevalence of overweight among children aged 0–59 months was 5% which is relatively higher when compared to neighboring countries such as Ethiopia and Burundi which in 2021 had less than 3% of under-fives with overweight/obesity [16] although the prevalence is much lower when compared to developed countries such as the United Arab Emirates where 21.5% of children are said to be overweight [17]. Maternal BMI was strongly associated with child overweight. The region where the mother lives was associated with child obesity with higher odds of overweight among Central, Western and Nairobi regions compared to North Eastern. In addition, urban residency was associated with overweight, although the association was weak. Other maternal factors showed no clear association. Among child-related factors, long breastfeeding period and less than 3-year-old children were associated with lower odds of childhood overweight whereas male gender was associated with higher odds of being overweight. 

Earlier studies [7,8] found that mothers who are experiencing overweight and obesity have increased odds of having overweight children, which was consistent with our findings. The results of the present study demonstrated that maternal nutritional status was also strongly associated with childhood nutritional status. Compared to maternal normal weight, overweight and obesity among mothers was associated with increased odds of children being overweight. However, the magnitude of the association of overweight and obesity among mothers with children experiencing overweight did not differ much. We also found that the prevalence of overweight children who were born from overweight and obese mothers was also about the same. Earlier studies have reported that high maternal pregnancy BMI is associated with higher odds of infant birth weight [18]. In addition, the association between maternal overnutrition and child overweight may be linked to genetic factors that may predispose the family to overweight [7]. On the other hand, the environment in which mothers and children are living may predispose them to the risk of developing overweight. Children are likely to be overweight if they reside in urban centers which give access to highly saturated fatty foods [19]. Additionally, living in urban centers provides limited access to green spaces, leading to limited physical activities, and thus a higher prevalence of overweight. Provision of green spaces in urban areas for the urban residents may act as a catalyst in promoting physical activity in the population [20].

The current study found a significantly higher prevalence of overweight children among urban residents (6.1%) compared to rural residents (3.9%). This confirms the effects of urbanization and urban lifestyle as observed in a previous study conducted in India [19], in tandem with a previous study conducted in Tehran, Iran, which showed that the industrialization of Tehran might have contributed to changes in lifestyle and food consumption, thus leading to a high prevalence of overweight in children under 5 years in Tehran, Iran [21]. In addition, we found more than 2-fold higher odds of childhood overweight in regions such as Nairobi and 1.9-fold higher odds in Central regions which are the regions that host the largest cities in Kenya. Families living in urban centers have easy access to high caloric cereals and fast foods when compared to those living in rural areas, thus a high prevalence of overweight. In addition, urban centers are associated with better economic status [22]. Therefore, the higher prevalence of overweight in urban centers is articulated to improved economic status in the cities and easy access to fast food merchandise. 

We found that parental attainment of higher education level was associated with increased odds of childhood overweight, although in the multivariate model, the association was not statistically significant. However, these results were in line with previous studies conducted in Kenya which found that maternal attainment of a higher level of education was associated with higher odds of childhood overweight [7,19]. This is supported by that fact that higher parental education increases parental possibilities of employment which in turn will boost the household income levels. In addition, highly educated and employed parents patently have busy work schedules which could make it easy for them to buy ready-made meals for their children rather than making home meals.

Regarding the wealth index, we found a higher prevalence of overweight from the richer and richest quintile, which was contrary to earlier findings [23]. Being in a richer and richest wealth quintile was significantly associated with increased odds of overweight. These findings were consistent to a study conducted in Bangladesh which found that households with the richest wealth index had higher odds of childhood overweight when compared to the poorest wealth index [24]. As earlier studies have indicated, family economic status may indirectly impact overweight through dietary intake. Wealthy families may invest in healthy food and less physical activities; much of the time could be spent on watching television. However, in the multivariable model adjusted for maternal and child-related factors, the association of wealth quintiles with childhood overweight was not clear. 

One earlier study stated that as opposed to stunting, childhood overweight has been associated with having no sibling [25]. We found that households with only one child had the highest prevalence of overweight as opposed to those with two or more children. This finding is consistent with an earlier study which stated that large family size is associated with lower odds of overweight and obesity [7]. However, in our study, the significant association was lost in the multivariate model. We found that male gender was associated with increased odds of being overweight as compared to females, which was consistent with other studies [26,27]. A longitudinal cohort study conducted in Vietnam and a cross-sectional study conducted in Kenya found that males were more likely to be overweight or obese when compared to females [7,28]. There is no clear mechanism for gender-based difference in childhood overweight [29]. However, some studies have found that gender differences in body composition, patterns of weight gain, hormone biology, and the susceptibility to certain social, ethnic, genetic, and environmental factors [30]. Furthermore, only few studies have explored the gender-based differences in childhood overweight and obesity.

We found that 12–23 months old children had higher odds of being overweight compared to 49–59 months old children, which was consistent with the findings of a study conducted in Uganda [27]. The situation results from adiposity at birth and high birth weight as reported in previous studies [31,32]. An observational prospective cohort conducted in Mexico found that increased size at birth and accelerated growth of new-borns in their first year increased the odds of overweight [33]. On the other hand, it will also depend on whether the children are breastfeeding or introduced to supplementary feeding in their first year. 

Breastfeeding is known to lower the risk of diarrhoea, respiratory illness, chronic diseases such as inflammatory bowel diseases and diabetes, and to significantly lower the risk of malnutrition [34]. These benefits are more significant in settings that are characterised by low standards of living, poverty, and challenges in maintaining a proficient level of nutrition and hygiene. Exclusive and prolonged breastfeeding reduces the odds of childhood overweight and mortality [35,36]. The current study found that the longer the period a child was breastfed, the lower the odds of being overweight. Compared to children breastfed for only twelve months, those breastfed for more than two years had lower odds of overweight. Our findings were consistent with earlier studies which showed that breastfeeding reduces the odds of childhood overweight [35,37]. Exclusive breastfeeding for at least six months and even continuing partial breastfeeding are the evidence-based standards to promote infant and child health globally. It is evident that the risk of all-cause mortality in infants are higher in partially and non-breastfed children when compared to those who are exclusively breastfed [38]. According to Kenya National Bureau of Statistics (KNBS), in Kenya 61% of children less than age six months are exclusively breastfed and over 81% of breastfed children aged six–nine months were introduced to early complementary foods in addition to being breastfed [4].

The evidence from this study calls for the designing of nutritional intervention programs targeting children aged between 0–59 months in Kenya. However, it is important to note that improving the Kenyan child’s nutritional status calls for multisectoral collaborations which may ease the magnitude of overweight. The generated evidence could also be vital in aiding Kenya as a country in its journey towards the achievement of the Sustainable Development Goal Two which calls for an end to hunger, achievement of food security and improved nutrition, and promoting sustainable agriculture [39], and the global nutrition targets 2025 for ‘no increase in childhood overweight’ [40]. 

### Limitations of the Study

The strength of this study includes the fact that it was conducted on a nationally representative sample. The results presented are population-representative and generalizable to similar low-and middle-income countries. To the best of our knowledge, this is one of the few studies of this kind in Kenya to analyze overweight among a unique population of children 0–59 months old. However, the study is limited to the fact that it used data which was collected in the year 2014. By the time the study was conducted, the Kenya demographic health survey data for 2022 had not yet been released to the public. The nutritional status of children may have changed in recent years. In addition, the study includes information of mothers aged 15–49 years excluding women who were ≥50 years old and were able to bear children. This calls for more studies that includes children from mothers of all medically possible childbearing ages. In relation to the classification of parents’ or guardians’ education level, the study took into consideration only formal education, leaving a gap in informal education attained by the parents or guardians, if any, that was unexplored. Nevertheless, the trend of overweight is increasing in most of low- and middle-income nations and Kenya may be following the same pattern. For this reason, there is still a need for more studies to be conducted using the most current data. In addition, given the nature of the study design, causal association could not be ascertained. 

## 5. Conclusions

We found a national prevalence of early childhood (0–59 months) overweight of 5% in Kenya, which was higher among male than among female children. The results suggested that maternal BMI was associated with childhood overweight, with a higher likelihood of overweight among the children of overweight and obese mothers. Urban residence and the region where the mothers live are associated with child obesity, with higher odds of overweight among those living in the Central, Western and Nairobi regions. In addition, breastfeeding for more than two years is a key factor associated with childhood overweight among Kenyan children. To tackle overnutrition issues in Kenya, there is a great need for public health interventions to target these maternal and child-related factors associated with childhood overweight. 

Childhood overweight is on the rise among Kenyan children. The increase is being witnessed at a moment when the country is still grappling with challenges of undernutrition. The findings of the current study provide important baseline information for stakeholders in planning for interventions to tackle maternal and childhood overweight for children aged 0-59 months. The study suggests the need to implement evidence-based maternal and child health policy and preventive strategies for childhood overweight. Consequently, developing effective strategies to tackle childhood overweight will require sufficient evidence on overweight childhood, so there is a need for more studies to provide current information on this issue. 

## Figures and Tables

**Table 1 ijerph-20-01413-t001:** Demographic characteristics of mother and child-related factors of the studied population.

Characteristics	*n* = 8318	Percentages
**Mother’s BMI**		
Underweight	705	8.5
Normal weight	5092	61.2
Overweight	1770	21.3
Obese	749	9.0
**Age of mother**		
15–19	403	4.8
20–24	1939	23.3
25–29	2559	30.8
30–34	1748	21.0
35–39	1108	13.3
40–49	560	6.7
**Place of residence**		
Rural	2868	34.5
Urban	5449	65.5
**Region**		
North Eastern	268	3.2
Coast	854	10.3
Eastern	1056	12.7
Central	754	9.1
Rift Valley	2410	29.0
Western	790	9.5
Nyanza	1012	12.2
Nairobi	1173	14.1
**Educational level of mother**		
No education	1006	12.1
Primary	4690	56.4
Secondary	1978	23.8
Higher	642	7.7
**Wealth index**		
Lowest	1983	23.8
Lower	1741	20.9
Middle	1561	18.8
Higher	1471	17.7
Highest	1560	18.8
**Partner’s educational level**		
No education	783	10.1
Primary	3965	51.0
Secondary	2152	27.7
Higher	868	11.2
**Total children ever born**		
1	1429	17.2
2	1887	22.7
3	1588	19.1
4	1111	13.4
5	735	8.8
≥6	1567	18.8
**Sex of child**		
Male	4188	50.4
Female	4129	49.6
**Child age**		
0-11 months	1663	20.0
12-23 months	1771	21.3
24-35 months	1679	20.2
36-47 months	1655	19.9
48-59 months	1549	18.6
**Months breastfeed**		
≤12 months	3175	38.8
12-24 months	4217	51.6
≥25 months	787	9.6
**Child’s BMI**		
≥+2sd	412	5.0
<+2sd	7904	95.0

**Table 2 ijerph-20-01413-t002:** Distribution of maternal demographic and child-related factors by child’s body mass index (BMI).

Characteristics	Totals (*n*)	Child’s BMI		*p*-Value
		Normal or Underweight	Overweight	
**Mother’s BMI**				<0.001
Underweight	705	696 (98.7)	9 (1.3)	
Normal weight	5092	4883 (95.9)	209 (4.1)	
Overweight	1771	1635 (92.3)	136 (7.7)	
Obese	749	690 (92.1)	59 (7.9)	
**Age of Mother**				<0.001
15–19	869	820 (94.4)	49 (5.6)	
20–24	4130	3928 (95.1)	202 (4.9)	
25–29	5287	5004 (94.6)	283 (5.4)	
30–34	3570	3408 (95.5)	162 (4.5)	
35–39	2267	2191 (96.6)	76 (3.4)	
40–49	1169	1137 (97.3)	32 (2.7)	
**Place of residence**				<0.001
Urban	5927	5566 (93.9)	361 (6.1)	
Rural	11,365	10,921 (96.1)	444 (3.9)	
**Region**				<0.001
coast	1775	1711 (96.4)	64 (3.6)	
North Eastern	557	510 (96.9)	17 (3.1)	
Eastern	2148	2058 (95.8)	90 (4.2)	
Central	1606	1495 (93.1)	111 (6.9)	
Rift Valley	5048	4841 (95.9)	207 (4.1)	
Western	2031	1958 (96.4)	73 (3.6)	
Nyanza	2448	2311 (94.4)	137 (5.6)	
Nairobi	1678	1572 (93.7)	106 (6.3)	
**Educational level of mother**				<0.001
No education	2058	2014 (97.9)	44 (2.1)	
Primary	9735	9364 (96.2)	371 (3.8)	
Secondary	4057	3798 (93.6)	259 (6.4)	
Higher	1441	1311 (91.0)	130 (9.0)	
**Wealth index**				<0.001
Lowest	4178	4070 (97.4)	108 (2.6)	
Lower	3631	3497 (96.3)	134 (3.7)	
Middle	3182	3033 (95.3)	149 (4.7)	
Higher	2970	2787 (93.8)	183 (6.2)	
Highest	3330	3100 (93.1)	230 (6.9)	
**Partner’s educational level**				<0.001
No education	783	763 (97.6)	20 (2.6)	
Primary	3965	3794 (95.7)	171 (4.3)	
Secondary	2152	2035 (94.6)	117 (5.4)	
Higher	867	792 (91.3)	75 (8.7)	
**Total children ever born**				<0.001
1	3116	2908 (93.3)	208 (6.7)	
2	4073	3861 (94.8)	212 (5.2)	
3	3213	3062 (95.3)	151 (4.7)	
4	2238	2154 (96.2)	84 (3.8)	
5	1494	1436 (96.1)	58 (3.9)	
≥6	3155	3065 (97.1)	90 (2.9)	
**Sex of child**				<0.001
Male	8764	8281(94.5)	483(5.5)	
Female	8528	8206(92.6)	322(3.8)	
**Child age**				<0.001
0–11 months	3445	3172 (92.1)	273 (7.9)	
12–23 months	3589	3373 (94.0)	216 (6.0)	
24–35 months	3536	3393 (96.0)	143 (4.0)	
36–47 months	3489	3374 (96.7)	115 (3.3)	
48–59 months	3232	3174 (98.2)	58 (1.8)	
**Months breastfed**				<0.001
≤12 months	3175	2972 (93.6)	203 (6.4)	
24 months	4217	4025 (95.4)	192 (4.6)	
≥25 months	787	774 (98.3)	13 (1.7)	

**Table 3 ijerph-20-01413-t003:** Association of maternal and child-related characteristics with child’s Overweight. Odds ratio (OR) and their 95% confidence intervals (CIs) from the logistic regression model.

Characteristics	OR, 95% CI		
	Model I	Model II	Model III
**Mother’s BMI**			
Underweight	0.29 (0.14–0.57)	0.30 (0.14–0.63)	0.30 (0.14–0.64)
Normal	1	1	1
Overweight	1.94 (1.56–2.43)	1.63 (1.27–2.08)	1.64 (1.28–2.11)
Obese	1.99 (1.47–2.68)	1.66 (1.18–2.34)	1.74 (1.22–2.47)
**Mother’s age**			
15–19	1	1	1
20–24	0.85 (0.62–1.17)	0.82 (0.45–1.50)	0.97 (0.52–1.79)
25–29	0.94 (0.69–1.28)	0.96 (0.52–1.79)	1.34 (0.71–2.51)
30–34	0.79 (0.57–1.09)	0.79 (0.40–1.55)	1.18 (0.60–2.35)
35–39	0.58 (0.40–0.83)	0.62 (0.30–1.30)	1.00 (0.47–2.12)
40–49	0.47 (0.30–0.74)	0.52 (0.22–1.25)	0.98 (0.40–2.38)
**Place of residence**			
Urban	1.60 (1.38–1.84)	1.14 (0.86–1.52)	1.19 (0.89–1.60)
Rural	1	1	1
**Region**			
Coast	0.85 (0.49–1.46)	2.17 (0.95–4.97)	2.27 (0.97–5.27)
North Eastern	1	1	1
Eastern	1.17 (0.84–1.62)	1.19 (0.70–2.04)	1.26 (0.73–2.16)
Central	1.99 (1.45–2.72)	1.88 (1.13–3.13)	1.86 (1.11–3.11)
Rift Valley	1.15 (0.86–1.53)	1.35 (0.85–2.15)	1.39 (0.87–2.22)
Western	1.81 (1.32–2.48)	1.74 (1.05–2.89)	1.75 (1.04–2.93)
Nyanza	1.00 (0.71–1.40)	1.33 (0.77–2.29)	1.36 (0.79–2.34)
Nairobi	1.57 (1.17–2.15)	1.95 (1.19–3.19)	2.02 (1.23–3.34)
**Education level of mother**			
No education	0.22 (0.15– 0.31)	0.47 (0.23–0.97)	0.54 (0.26–1.13)
Primary	0.40 (0.32–0.49)	0.56 (0.36–0.88)	0.65 (0.41–1.02)
Secondary	0.69 (0.55–0.85)	0.86 (0.58–1.28)	0.95 (0.64–1.42)
Higher	1	1	1
**Wealth index**			
Lowest	1	1	1
Lower	1.44 (1.12–1.87)	1.00 (.67–1.50)	1.05 (0.70–1.57)
Middle	1.85 (1.44–2.38)	1.12 (0.75–1.68)	1.14 (0.76–1.71)
Higher	2.46 (1.93–3.14)	1.00 (0.65–1.53)	1.00 (0.65–1.55)
Highest	2.78 (2.21–3.51)	0.88 (0.54–1.45)	0.84 (0.51–1.39)
**Partner’s educational level**			
No education	0.28 (0.17–0.46)	0.80 (0.40–1.60)	0.78 (0.39–1.56)
Primary	0.47 (0.36–0.63)	0.89 (0.61–1.31)	0.89 (0.60–1.31)
Secondary	0.60 (0.45–0.82)	0.83 (0.58–1.19)	0.82 (0.57–1.17)
Highest	1	1	1
**Total children ever born**			
1	1	1	1
2	0.77 (0.63–0.94)	0.70 (0.51–0.97)	0.78 (0.56–1.08)
3	0.69 (0.56–0.86)	0.71 (0.49–1.02)	0.72 (0.49–1.08)
4	0.55 (0.42–0.71)	0.87 (0.52–1.10)	0.86 (0.55–1.35)
5	0.56 (0.42–0.76)	0.87 (0.52–1.46)	0.91 (0.54–1.53)
≥6	0.42 (0.32–0.53)	0.64 (0.37–1.10)	0.59 (0.34–1.04)
**Sex of child**			
Male	1.49 (1.29–1.72)		1.53 (1.24–1.90)
Female	1		1
**Child age**			
0–11 months	4.69 (3.52–6.25)		0.88 (0.61–1.26)
12–23 months	3.49 (2.60–4.68)		0.50 (0.33–0.74)
24–35 months	2.30 (1.69–3.14)		0.45 (0.30–0.67)
36–47 months	1.85 (1.35–2.55)		0.25 (0.16–0.40)
48–59 months	1		1
**Months breastfed**			
≤12	1		1
12-24	0.70 (0.57–0.86)		0.97 (0.72–1.31)
≥25	0.25 (0.15–0.44)		0.51 (0.28–0.95)

Model I: Crude association. Model II: Multivariable model adjusted for maternal factors. Model III: Multivariable model (Model II + child-related factors).

## Data Availability

Not applicable.

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
