# Peer review of "The Association of Maternal BMI with Overweight among Children Aged 0–59 Months in Kenya: A Nationwide Cross-Sectional Study"

_ijerph, 2023, doi:10.3390/ijerph20021413_

Round 1

Reviewer 1 Report (Previous Reviewer 1)

The authors addressed most of the comments with regards to writing style and methodology description. However, the authors did not address the statistical analysis concerns of using the smallest groups in breastfeeding duration and maternal weight status as comparison groups. For breastfeeding duration, the use of >=25 months as a comparison group is unconventional and is perhaps the least ideal given that this group makes less than 10% of the entire study population. Similarly, the authors used underweight mothers (making up 8% of the study population) as the comparison group, where normal weight mothers are the obvious choice. The overlap in odds ratios for child's overweight between important groups like Normal vs overweight mothers for maternal weight and <12 and 12-24 months for breastfeeding is still not discussed, which is problematic as those findings are in conflict with the conclusions that the authors offer. Perhaps the study is underpowered to detect those statistical differences, however this cannot be avoided by using an unconventional comparison group in order to produce desired results. 

Author Response

Please find attached the response.

Reviewer 2 Report (Previous Reviewer 3)

The authors have improved the paper.

Please, insert a heading "Limitations of the Study" in the last paragraph of the discussion before listing the limitations. Some typos were introduced while writing the limitations of the study.

The references were still not listed according to the journal style

Author Response

Please find attached the response.

This manuscript is a resubmission of an earlier submission. The following is a list of the peer review reports and author responses from that submission.

Round 1

Reviewer 1 Report

 1 - Line 83: Please describe method of data collection, including how height and weight were measured by survey workers. Maybe include link to online information on the entire project?

2- Line 87: Were the height and weight measurements obtained by survey workers or self reported by parents? this should be stated in the methods.

3- Line 93: "N" should be used to indicate population size. "n" should be used to indicate sample size.  Is 8316 the number of mothers in the sample, or children, or mother-child pairs?

4 - Line 157: 22.7% is not a majority

5 - Line 158: Why is "at least" being used to describe exact percentages, do authors believe these numbers under-estimate the true percentages? if so, why?

6 - Table 1 Ranges should be indicated for Children ages and Months of breastfeeding

7 - Line 241: Why is the comparison being made between underweight vs overweight instead of normal vs overweight?

8 - Lines 215, 302: "Becoming" here implies future change, as this is a cross-sectional and not longitudinal study, "being" should be used instead.

9 - Line 304: The use of the word "articulated" here is unclear

10 - The authors should discuss the finding of no significant difference in overweight risk between breast feeding for less than 12 months and 12-24 months. Null results should also be discussed when examining significant between-group differences. Additionally, there was no mention of mothers who, for one reason or another, did not breastfeed, were those included in the 12 months group? If so this should be stated in the relevant parts of the  methods/discussion sections.

Reviewer 2 Report

The authors investigate the relationship between children’s overweight and mothers’ BMI and identify potential risk factors. They find mother’s BMI, and urban residence, geographical location, length of breastfeeding and child’s gender are associated to children’s overweight status. They conclude that there is a need to develop public health interventions to tackle maternal and child related problems.

The paper is straightforward, and the issue is worth being investigated. Methods are appropriate and adequately described. The results can be useful. My main concern is that significance parameters have not been included either in the Chi-square and the logit regressions’ results. In addition, the paper’s aim and discussion need to be improved. I would also suggest somewhat improving the introduction’s structure.

Below are several comments and suggestions that should be addressed.

·       The paper’s aim needs to be better outlined. The paper’s objective encompasses more than what is set out in lines 68-71. It needs to be more comprehensive, so I would suggest further elaborating it.

·       Line 100: The range corresponding to underweight is not completely correct. It must be a typo, but please correct it.

·       Table 1. Classification of variable “Children ages”: It is not at all clear from the table the authors mean age ranges. I would suggest including the age ranges.

·       Tables 1 and 2: I would suggest being consistent and using the exact same names for each of the variables. The same applies to Table 3.

·       Chi-square tests’ results have not been included.

·       Table 3: I would suggest including the significance of the relationship in the Table (with asterisks for the different levels of significance). I think significance parameters are not included anywhere in the paper. It would be much easier to understand the results.

·       Line 216: Is “reversed” the correct word here? I would suggest checking this.

·       The discussion needs to be better worked out. There are a number of blunt statements which need to be nuanced. It is at times a bit repetitive. The structure needs to be improved, so I would suggest organizing it somewhat better.

Reviewer 3 Report

The manuscript is interesting as it bothers on an emerging global public health issue, particularly in children.  However, there are issues to addressed as listed below

1.      Line 8: “…. steadily affecting many countries”? Do you mean …. prevalent in many countries?

2.      Lines 11-12: “This cross-sectional study used 2014 Kenya Demographic and Health survey which is a national representative data” Not clear. Please reword the sentence for clarity. Do you mean “ This cross-sectional study used Kenya’s 2014 demographic and health survey data….”

3.      Line 12: No need for providing the age range again since it has been stated in line 11

4.      Lines 26-33: Please provide some background information to enable the readership differentiate between overweight and obesity.

5.      Line 45: Not sure of what the authors’ meant by “psychological health” in this context. Are you referring to metal alertness or personality problem such as (self-induced) inferiority complex probably due to reduced social acceptance? What exactly do you mean here because I do not see a direct correlation between overweight/obesity and mental illness?

6.      Introduction: Besides nutrition, urbanization and sedentary life style, could genetic factors play any role in child/maternal obesity or overweight? Such background information may be helpful in the introduction.

7.      Lines 75-81: The use of 2014 KDHS data in 2022 for this study warrants being listed as a limitation to this study.

8.      Lines 81-84: How did the authors select the clusters and households surveyed? Which sampling technique/method was used in the selection?

9.      Lines 102-103: Is conception and child bearing medically impossible in ALL women aged 50-54 year? If no, why was this age category excluded in the study design? If there is no valid justification for the exclusion, then consider this as another limitation of this study.

10.  Lines 109-110: Replace “measured” with “determined”; “no education” with “no formal education”; “higher” with “tertiary”. Apply the changes in the results section and all though the manuscript.

11.  Ethical statement: Beyond statement of informed consent, an ethical approval is critically important for this study. The authors collected data (height, weight, etc.) from participants. This makes ethical clearance from a relevant body imperative for this study

12.  Tables 1, 2 and 3: Children age categorization listed in Lines 123-125 are not consistent with that listed in the Tables 1, 2 and 3.

13.  Lines 229-230: How does the 5% national prevalence found compare with that of other African and non-African countries?

Round 2

Reviewer 2 Report

The paper has been improved and most of the comments have been addressed satisfactorily. I would still suggest being consistent and using the exact same names for each of the variables in Tables 1, 2 and 3. I am aware the Chi-square test results are presented in table 2, but I would still suggest noting that the results are included in Table 2, either in the text or in the table’s title (or in both). It is true the significance of the estimates can be understood from the confidence interval provided for the estimates, but I still think it is clearer when the p-value significance is also included. But this of course is up to the authors.

Reviewer 3 Report

1.

The authors have greatly improved the manuscript but I disagree with then on the categorization of educational level all through the manuscript.

No formal education does not in any way translate to “no education”.  A respondent may have no formal education but may be sophisticated educated in other areas. For instance, an Islamic scholar may acquire Islamic education but not Western education. Thant does not make him/her not educated or “no education”.  Others may be educated in one skill/craft or the other but may have no Western education. How can the authors classify such respondents as not educated or no education?

If the authors choose to stick to the DHS standard on categorization of educational levels, then the study is limited in this area and should be mentioned as another limitation of the study.    
